# Effect of Bacteria Inoculation on Colonization of Roots by *Tuber melanosporum* and Growth of *Quercus ilex* Seedlings

**DOI:** 10.3390/plants13020224

**Published:** 2024-01-13

**Authors:** Veronica Giorgi, Antonella Amicucci, Lucia Landi, Ivan Castelli, Gianfranco Romanazzi, Cristiano Peroni, Bianca Ranocchi, Alessandra Zambonelli, Davide Neri

**Affiliations:** 1Department of Agricultural, Food and Environmental Sciences, Polytechnic University of Marche, 60131 Ancona, Italy; v.giorgi@staff.univpm.it (V.G.); l.landi@staff.univpm.it (L.L.); i.castelli@staff.univpm.it (I.C.); g.romanazzi@univpm.it (G.R.); d.neri@staff.univpm.it (D.N.); 2Department of Biomolecular Sciences, University of Urbino “Carlo Bo”, 61029 Urbino, Italy; b.ranocchi@campus.uniurb.it; 3Agenzia per l’Innovazione nel Settore Agroalimentare e della Pesca “Marche Agricoltura Pesca”, AMAP, 60027 Osimo, Italy; studioverdeperoni@libero.it; 4Department of Agricultural and Food Sciences, University of Bologna, 40127 Bologna, Italy; alessandr.zambonelli@unibo.it

**Keywords:** bacteria, *Tuber melanosporum*, fibrous roots, pioneer roots, *Bradyrhizobium*, *Pseudomonas*, ectomycorrhizae, *Quercus ilex*, mycorrhization

## Abstract

*Tuber melanosporum* is an ascomycete that forms ectomycorrhizal (ECM) symbioses with a wide range of host plants, producing edible fruiting bodies with high economic value. The quality of seedlings in the early symbiotic stage is important for successful truffle cultivation. Numerous bacterial species have been reported to take part in the truffle biological cycle and influence the establishment of roots symbiosis in plant hosts and the development of the carpophore. In this work, three different bacteria formulations were co-inoculated in *Quercus ilex* L. seedlings two months after *T. melanosporum* inoculation. At four months of bacterial application, the *T. melanosporum* ECM root tip rate of colonization and bacterial presence were assessed using both morphological and molecular techniques. A 2.5-fold increase in ECM colonization rate was found in the presence of *Pseudomonas* sp. compared to the seedlings inoculated only with *T. melanosporum*. The same treatment caused reduced plant growth either for the aerial and root part. Meanwhile, the ECM colonization combined with *Bradyrhizobium* sp. and *Pseudomonas* sp. + *Bradyrhizobium* sp. reduced the relative density of fibrous roots (nutrient absorption). Our work suggests that the role of bacteria in the early symbiotic stages of ECM colonization involves both the mycorrhizal symbiosis rate and plant root development processes, both essential for improve the quality of truffle-inoculated seedlings produced in commercial nurseries.

## 1. Introduction

The fungal species in the genus *Tuber* (Ascomycota, Tuberaceae) are able to produce hypogenous edible fruiting bodies, known as “true truffles” [1]. Some species have high economic value due to the gastronomical characteristics of their fruiting body, like *Tuber melanosporum* Vittad., producing edible ascomata known as ‘black truffles’. *Tuber* species establish an ectomycorrhizal (ECM) mutualistic symbiotic relationship with the roots of forest plants and benefit the growth of the host plants [2]. Thus, *Tuber* spp. play a great role in vegetation and forest ecosystems by influencing water and nutrition absorption, plant growth, and resistance to pathogens [3]. Studying valuable species of the genus *Tuber* is important not only to support their production but also to understand their little-known biology. This knowledge can be used to support rural development programs and improve marginal or uncultivated agricultural areas.

Historically, truffles started to be cultivated in the XIX century and, starting from the 1960s, improvements in agricultural techniques were studied and applied [4], meaning that natural production in forests was greatly replaced by truffle production in cultivated orchards [5]. At the same time, climate change and anthropization have also strongly influenced the natural production of tubers in forests [6]. As a result, nowadays, most of the truffles harvested worldwide are produced in orchards with seedlings previously inoculated in controlled conditions in a nursery. Different species can be used as host plants; the most common are oak species (*Quercus* spp.) and, among them, *Quercus ilex* is one of the most widespread species in the Mediterranean area [7]. In this scenario, there is considerable interest in improving truffle cultivation, understanding plant–fungus interactions, and maximizing production through new strategies.

The success of truffle production is strongly influenced by the quality of seedlings and their inoculation. The mycorrhized seedling quality must be intended for both organisms: the plant and the fungus. The Italian Minister of Agriculture and Forestry sets key aspects for plant evaluation, such as a properly developed root system, an adequate shoot-to-root ratio, a good height/diameter ratio at the collar, and the absence of damage and pathogens [8,9]. Meanwhile, inoculated seedlings must guarantee the absence of undesirable truffle species and an adequate mycorrhizal colonization level, preferably certified [10]. Compliance with these conditions promotes initial plant growth and facilitates subsequent infections by enhancing the meeting of root tips with spores [11].

In this context, it is necessary to analyze the roots of the seedlings to ensure good performance in the orchard. Moreover, it is known that *T. melanosporum* colonizes specific roots in the host plants [12,13]. The fungal mycelium only grows on the fibrous roots, which are short, thin, and numerous, and, in the presence of the fungus, become darker in colour [14]. Pioneer roots are another type of root that plays an important role in the evaluation of the root system. These roots are longer, thicker, and lighter in colour, and help in exploring new areas. The ratio between these two types of roots is an indicator of root architecture and the system’s ability to establish a symbiosis with the ECM fungi [15], and it can be modified via environmental factors and the presence of mycorrhizal symbiosis [16,17].

Several ways to improve truffle production have been studied: management practices at the greenhouse stage [18,19]; standardized procedures to evaluate seedling quality for customers benefit [10,20,21,22]; and growth conditions promoting *T. melanosporum* proliferation in the field [23,24,25,26].

The use of bacteria has been also proposed recently to improve the truffle biological cycle because bacteria take part in the interaction between fungus and plant [27]. They play a role in influencing the aroma, [28] promoting plant growth, protecting against pathogens, and nitrogen fixation [29]. Among cultivation practices, it has been shown that combined inoculation of ECM and plant-growth-promoting rhizobacteria (PGPR) is an effective strategy for improving the quality of seedlings, increasing plant survival, enhancing mycorrhizal symbiosis establishment, and promoting plant growth [30,31].

It has been observed that the truffle species analyzed so far are colonized by bacterial communities primarily composed of Alphaproteobacteria, Bacteroidetes, Firmicutes, and Actinobacteria [32,33,34]. It was found that *Bradyrhizobium* and *Pseudomonas* were the most frequently detected genera in the bacterial communities associated with *Tuber* species, regardless of whether the bacteria were detected directly from total ascocarp DNA or isolated from mycelial cultures [35]. It has been reported that *Bradyrhizobium* is the most representative species among those identified through molecular detection, while *Pseudomonas* are the most representative species among those cultivable [35].

Interestingly, studies have shown that *Pseudomonas fluorescens* can enhance the symbiotic relationship between mycorrhizal fungi and host plants, and can increase their colonization, resulting in a positive effect on plant growth [30,36,37,38]. Furthermore, the Bradyrhizobia could also play a key role in the biological cycle of the truffle. They are known as symbiotes of legumes, particularly soybean plants. These bacteria can fix nitrogen, promote plant growth, and increase oxygen production, which is essential for the survival of various organisms [39,40]. In addition, in Barbieri et al. (2010), *Bradyrhizobium* spp. were detected inside *Tuber magnatum* ascocarps, and it was assumed that they could play a potential role in fungal growth or nutrition during ascocarp development [41].

Due to the results reported so far and their suggested potential, this study examined the effects of two representative *Pseudomonas* sp. and *Bradyrhizobium* sp. and their combination on the stimulation of the mycorrhization process of EMC and on the quality of *Q. ilex* seedlings and root development.

## 2. Results

### 2.1. Evaluation of the Identity and the Presence of Inoculated Bacterial Strains

In the preparatory phase, the identity of the cultured bacteria in liquid or solid media was confirmed through RFLP analysis of the 16S region, in all steps of preparation, as described in Materials and Methods. In Figure 1, the identification of *Bradyrhizobium* through RFLP of the 16S region is shown.

The permanence of the inoculated bacteria was verified in root samples using PCR with specific primers. This method is effective in achieving accurate results from a complex matrix in which many microorganisms are present. The results are shown in Figure 2.

### 2.2. Mycorrhizae Analysis

The results showed that the presence of mycorrhizae had different effects on plant growth. The identification of ectomycorrhizae was carried out via optical microscope observations, as shown in Figure 3. Further confirmation of the identity also emerged from the molecular data. A qualitative PCR was performed on mycorrhizae samples of apical tips, and the expected band of 438 bp of *T. melanosporum* was observed in all the samples, except the negative control (Figure 4).

### 2.3. Plant and Root Architecture Analysis

There was no significant effect on the shoot/root ratio caused by the treatments. However, treatment 3 resulted in a higher shoot growth rate compared to treatment M2 (Figure 5).

Generally, the presence of mycorrhizae and bacteria influenced root growth and architecture. The root biomass of pioneer and fibrous roots was statistically different in non-mycorrhized and mycorrhized plants, with a higher biomass of fibrous roots in the non-mycorrhized plants (Figure 6).

Comparing the effects of different bacterial species, the partitioning of root biomass differed if the bacteria were combined with *T. melanosporum* inoculum; in this case, bacterial inoculum 1 and 3 caused a decrease in fibrous root biomass, while inoculum 2 (*Pseudomonas* sp.) showed a higher fibrous root biomass similar to the control group (which had no mycorrhization and no bacteria) (Figure 7).

### 2.4. Mycorrhization Degree

Regarding the *Q. ilex* root colonization, all the seedlings inoculated with *T. melanosporum* were colonized. A non-significant cross-contamination with the *T. melanosporum* inoculum was detected in the treatment 0, 1, 2, 3. Overall, at four months from bacterial treatment, the average percentage of *T. melanosporum* colonization increase rose from 2.1-fold, for the M3 mix, to 2.5-fold for the M2 mix compared to the M0 treatment without bacterial inoculum (Figure 8A). This indicates a general enhancement in *T. melanosporum* establishments in oak roots following treatment with bacterial formulations. However, the most significant effects were caused by the M2 bacterial mix based on *Pseudomonas* sp. (Figure 8A). Interestingly, the root tips proximal to the shoot junction showed a higher percentage of mycorrhization than the distal root tips (originating from the end of the tap root). This was evident for the M2 treatment, which increased the *T. melanosporum* root colonization 3.8-fold compared to M0 (Figure 8B). Meanwhile, no significant differences in *T*. *melanosporum* colonization were found between M0 and the other treatments (M1, M2, M3) on the distal roots (Figure 8C).

## 3. Discussion

Our study found that six months after *T. melanosporum* inoculation and four months from bacteria co-inoculation, the ECM symbiosis with *Q. ilex* plants was improved by the presence of the bacteria, compared to the plants inoculated only with *T. melanosporum*. An excellent affinity between *Q. ilex* and *T. melanosporum* was previously indicated at one year from mycorrhizal inoculation, ranging from 10–20% [42] to 50% mycorrhized root tips [24]. Our data reveal that all seedlings co-inoculated with bacteria formulations tended to have better mycorrhization rates starting from the early symbiosis phase than those inoculated only with *T. melanosporum*. In detail, significant promotion was detected using the M2 formulation based on *Pseudomonas* sp. Recently, a comprehensive study of the microbiomes in truffle orchards showed that the *Pseudomonas* and *Bradyrhizobium* genera, which are known as “helper” bacteria, were present in high numbers [43,44]. Previous works indicated that these bacteria are able to promote root mycorrhization [36] and play a key role in establishing plant–fungal symbiosis [45]. They stimulate mycelial growth, affect the symbiotic relationship between the host plant and fungi, and reduce the effects of environmental stress on the mycelium [36,46]. A significant increase in *T. melanosporum* colonization in *Pinus halepensis* Mill. roots was observed in the presence of *P. fluorescens* [30]. Recently, the efficacy of bacteria in promoting *T. melanosporum* root colonization was assessed in *Quercus faginea* [31]. The authors showed that *P. fluorescens* increased ECM colonization by about 10% compared to the control (from 26.8 to 35.1%) after one year of co-inoculation with both bacteria and mycorrhizal fungi. Our work explores for the first time the role of these bacteria groups in the early phase of ECM symbiosis between *Q. ilex* and *T. melanosporum*. In this regard, the analysis of ECM symbiosis with proximal roots (close to the root collar) and distal roots (close to the end of the tap root) revealed that *T. melanosporum* displayed the highest colonization on proximal roots, mainly for the M2 treatment. An explanation could be related to the delayed development of the distal roots compared to the proximal ones; they therefore had more opportunities for *T. melanosporum*–root interaction.

However, the different results observed after the inoculation of different bacterial strains or their combination suggest that their presence does not necessarily increase the ECM–root-tip symbiosis. In fact, the combined use of *Bradyrhizobium* sp. and *Pseudomonas* sp. did not show any effect in promoting ECM root colonization. Our results suggest that *Bradyrhizobium* sp. may reduce the positive action of *Pseudomonas* sp. The efficacy of certain *Bradyrhizobium* species in growth and nitrogen fixation in association with arbuscular and ectomycorrhizal fungi has been observed in different plant–mycorrhiza symbiosis [47,48]. The studies demonstrate that these bacteria have a beneficial influence on plant growth. However, it is unclear whether species of the genus *Bradyrhizobium* can contribute to the increase in mycorrhizal symbiosis or use other strategies to support plant growth.

It was interesting to investigate the combined ability of *T. melanosporum* and the inoculated bacteria to affect the seedlings’ development. In the experimental trials, a similar shoot/root biomass ratio was found, indicating a similar proportion of resource partitioning between aerial and underground parts; however, seedlings co-inoculated with *Pseudomonas* sp. and *T. melanosporum* (M2) produced the smallest plants but with the highest presence of fibrous roots. Splivallo et al. [49] demonstrated that *T. melanosporum* can influence root morphology, stimulating lateral root formation and increasing branching through metabolite production. Mycorrhizal symbiosis has the same function for the plant as the fibrous roots, that is, to absorb water and nutrients. Fibrous roots and mycorrhizal symbiosis both require carbon from the plant for their development [50]. Thus, under the M2 treatment, the highly mycorrhized root system offsets the reduction in total biomass. A modification in the root architecture has been reported in a few other studies, in symbiosis between *T. melanosporum* and different plants [16,17,30]. The competitive success of the plant and the fungus depends on a well-branched root system rich in fine roots, which, on the one hand, ensures access to resources for the plant, and on the other, is the fundamental substratum for the fungus. Furthermore, the high presence of fine roots facilitates the mycelium belowground proliferation [51]. Although the presence of ectomycorrhizal roots is not a guarantee for ascocarp formation [52], their widespread presence is a good indicator of fructification [53].

The co-inoculation of bacteria and *T. melanosporum* significantly changes the amount of fibrous and pioneer roots. Interestingly, we observed a specific response of the fibrous root system (fibrous/total roots) to inoculation by different bacterial mixes: M1 and M3 (both containing *Bradyrhizobium* sp.) resulted in a smaller fibrous root rate, while M0 and M2 had a similar amount of fibrous roots compared to non-mycorrhizal ones. The M2 treatment also led to a higher ECM colonization, emphasizing that bacteria can have a great impact on both plant growth and mycorrhizal association.

## 4. Materials and Methods

### 4.1. Seedling Cultivation

The experimental trial was conducted in the forest nursery of the Agency for Innovation in the Agri-Food and Fisheries Sector of the Marche Region (AMAP), Amandola, AP, Central Italy. *Q. ilex* L. plants obtained from germinated seeds were transplanted in pots (volume 450 cl from ACUDAM, Lleida, Spain) after eight weeks of germination. The tap root was cut at 6–7 cm to promote lateral root development, and leaves were pruned by cutting 1/3 of their surface to rebalance the root/shoot ratio. A substrate composed of 1/3 of peat, 1/3 of natural soil, 2/9 of vermiculite, 1/9 of perlite with slow-release fertilizer (1 g per pot of OSMOCOTE 6M 19.6.11) and calcium carbonate to adjust pH at 7.8, was used. This substrate combination was shown to be the most efficient for the pot-plant species in the preliminary trials. The seedlings were grown in greenhouse conditions with day temperature of 29 °C, 30–40% humidity, and irrigation twice a week.

### 4.2. T. melanosporum Inoculation

Seedlings were inoculated during their transplantation in pots, with *T. melanosporum* spores obtained from ripe fresh truffles harvested in the Marche region, according to the procedure described by Iotti et al. [54]. The truffles were washed, cleaned of impurities and residual soil materials, crushed, and then dehydrated. The resulting powder was mixed with sand to obtain the inoculum to be placed in contact with the root. An equivalent amount of 2 g of fresh truffle was used per pot.

### 4.3. Bacterial Inocula Preparation

Two months after truffle inoculation, the bacterial cultures of *Bradyrhizobium* sp. and *Pseudomonas* sp. were added to the pots.

Liquid culture of *Bradyrhizobium* in YM (0.65 g/L K₂HPO₄ × 3H_2_O, 0.2 g/L MgSO_4_, 0.1 g/L NaCl, 10 g/L mannitol (C_6_H_14_O_6_), 0.4 g/L yeast extract) was obtained from the cultures grown on plates with solid YM with 15 g/L agar. For mixture 1, the *Bradyrhizobium* sp. bacterial strain was grown in YM liquid medium, pH 6.8–7, at 28 °C in a circular shaker for about seven days until reaching a total of 320 × 10^6^ CFU/mL (colony-forming unit), which equates to 153.6 × 10^8^ cells added to each seedling. The CFUs were calculated by creating a curve measuring the OD against the CFU counts in a plated serial dilution of the bacteria.

Liquid culture of *Pseudomonas* was prepared from colonies grown on solid TSA (tryptic soy agar, 40 g/L) plates inoculated in TSB (tryptic soy broth, 30 g/L) medium. For mixture 2, *Pseudomonas* sp. was cultivated in sterilized liquid TSB medium at 28 °C in a circular shaker for about two days until reaching a total of 797 × 10^6^ CFU/mL, which equates to 153 × 10^9^ cells added to each seedling.

For mixture 3, the 398.5 × 10^6^ *Pseudomonas* sp. cells were mixed with 160 × 10^6^ *Bradyrhizobium* sp. cells and added to each seedling. The mixtures were diluted with water before being added to the seedlings. To optimize the specific growth of *Pseudomonas* and avoid contamination, some growth stages were achieved using cetrimide agar medium (15.0 g/L agar, 0.3 g/L CTAB, 20.0 g/L gelatin peptone, 1.4 g/L MgCl_2_, 10.0 g/L K_2_SO_4_), a selective medium for *Pseudomonas*.

### 4.4. Experimental Setup

The three bacterial inocula were added to both mycorrhized and non-mycorrhized plants: 1. *Bradyrhizobium* sp., 2. *Pseudomonas* sp., and 3. *Pseudomonas* sp. + *Bradyrhizobium* sp. Therefore, the plants inoculated with both, *T. melanosporum* and bacteria strains were indicated as M1, M2, and M3. The 0 corresponds to control plants without both *T. melanosporum* and bacterial inocula, while M0 corresponds to plants inoculated only with *T. melanosporum.*

The eight treatments were replicated eight times, leading to a total of 64 seedlings. After four months from bacterial treatments, four plants per treatment were used to detect the presence of the inoculated bacterial species, while the biometric measures, root architecture, and mycorrhizal colonization were performed on the other four plants per treatment.

### 4.5. Molecular Identification of Bacterial Species

The amplification of the 16S ribosomal gene region was performed directly from picked bacterial colonies or diluted apical mycorrhizal tips. The reaction mixture, containing UP Forward and UP Reverse, and the thermocycling (Simply one, ThermoFisher, Waltham, Massachusetts, USA) setting, were performed according to Amicucci et al. [55].

The results of the amplification were visualized using 2% agarose gel in 0.5 X TBE (Tris-borate-EDTA) buffer and 5 µL of 10,000 X in water GelRed (Biotium, Landing Parkway, Fremont, CA). The molecular marker 100 bp DNA Ladder (New England Biolabs, Ipswich, Massachusetts, USA) was used. The images were acquired with Gel Doc 2000 system (Bio-Rad, Hercules, California, USA) using a UV transilluminator.

Following the manufacturer’s instructions, the amplified 16S region was digested with the restriction endonuclease *Rsa*I (New England Biolabs, Ipswich, Massachusetts, USA) [56] and incubated for 1 h at 37 °C. The digestion result was visualized using 2.5% agarose gel in 0.5 X TBE buffer.

The *Pseudomonas* sp. identification via PCR was performed using primers designed on the 16S gene, in a buffer containing 100 mM Tris-HCl with pH 9.0, 500 mM KCl, 1% Triton X-100, 15 mM MgCl_2_, with *Pse*I primers (5′-CTACGGGAGGCAGCAGTGG-3′) and *Pse*II (5′-TCGGTAACGTCAAAACAGCAAAGT-3′) [57], dNTPs 200 µM, and *Taq* DNA polymerase (Takara) 0.04 U/µL.

The amplification protocol was carried out in the thermocycler (Simply one, ThermoFisher, Waltham, Massachusetts, USA ), which consisted of an initial denaturation at 95 °C for 10 min, followed by 30 cycles consisting of a denaturation at 94 °C for 30 s, annealing at 62 °C for 15 s, extension at 72 °C for 30 s, and a final extension step at 72 °C for 7 min.

The amplified samples, of expected 150 bp size, were visualized on 2% agarose gel, in 0.5 X TBE, stained with Midori Green Dye (Nippon Genetics, Düren, Germany), and a molecular marker 100 bp DNA ladder was used. For *Bradyrhizobium* sp., the specific amplification reaction used primers designed on an Hsp70 bacterial gene in a buffer containing 10 mM Tris-HCl with pH 9.0, 50 mM KCl, 1% Triton X-100, and 1.5 mM MgCl_2_, *Brdnak*f primers (5′-TTCGACATCGACGCSAACGG-3′) and *Brdnak*r primers (5′-GCCTGCTGCKTGTACATGGC-3′) [58], dNTP 200 µM, and *Taq* polymerase (Takara) 0.04 U/µL. The thermocycling conditions consisted of an initial denaturation at 95 °C for 10 min, followed by 30 cycles of denaturation at 95 °C for 15 s, annealing at 58 °C for 15 s, extension at 72 °C for 20 s, and a final extension step at 72 °C for 7 min. The amplified samples, of expected 474 bp size, were visualized on 1.5% agarose gel.

### 4.6. Morphological and Molecular Identification of T. melanosporum Root Colonization

The presence and the structure of *T. melanosporum* mycorrhizae were first verified by checking specific morphological traits as the mantle anatomy and cystidia, using a Nikon microscope Eclipse TE 2000-E and a Nikon DXM1200F digital camera. This was performed according to the procedure outlined by Zambonelli et al. [59].

The morphological identification of *T. melanosporum* mycorrhizae was confirmed via molecular analyses on 3–10 randomly selected mycorrhizae per plant. The mycorrhizae were removed with tweezers and stored in 500 mL of physiologic solution (0.9% NaCl) at −20 °C. The mycorrhizae were then disrupted and homogenized in NaCl solution with a sterile pestle. Serial dilutions were prepared for each sample (1:10, 1:100, 1:1000, 1:10,000). From these dilutions, 1 µL was taken as a sample for the ITS region amplification.

Amplifications of the ITS region were performed according to Amicucci et al. [56], with the addition of BSA 0.4 mg/µL. The amplification protocol included an initial denaturation at 94 °C for 5 min, followed by 35 cycles of denaturation at 94 °C for 20 s, annealing at 58 °C for 15 s, extension at 72 °C for 40 s, and a final extension step at 72 °C for 7 min [56]. Samples were then visualized on 1% agarose gel in 0.5 X TBE buffer and stained with Midori Green Dye (Nippon Genetics).

Specific amplification was then performed in buffer containing 10 mM Tris-HCl pH 8.3, 50 mM KCl, 1.5 mM MgCl_2_, 0.4 µM of specific primers ITSML Forward (5′-TGGCCATGTGTCAGATTTAGTA-3′) and ITS4LNG Reverse (5′-TGATATGCTTAAGTTCAGCGGG-3′) [60], dNTPs 200 µM, *Taq* polymerase (Takara Bio, Kusatsu, Japan) 0.04 U/µL, and BSA for four samples and 100-fold more concentrated BSA for the remaining samples. DNA was added in serial dilutions of 10. Thermocycling was set up according to Paolocci et al. [60]. Samples were evaluated on 1.4% agarose gel stained with Midori Green Dye (Nippon Genetics).

### 4.7. Effects of Bacterial Formulations on Quercus Ilex Development and Plant Root T. melanosporum Colonization

The investigation was performed on four plants. The root system of each plant was split up longitudinally; half was used for architectural analyses of the root system, and the other half was used to evaluate the *T. melanosporum* ECM root tip colonization.

#### 4.7.1. Plant Measurements

Plants’ shoot growth was measured, then the plant roots were gently washed out of the soil to avoid any damage to small roots. To prevent the loss of small roots, the soil and roots were soaked in water for 30 min, and then the roots were gently washed under running water using a 0.21 mm sieve to collect the root fragments.

For the architectural analysis, the root system was divided into “fibrous roots” (approximately less than 2 mm in diameter) and “pioneer roots” (diameter greater than 2 mm) [61,62]. All plant parts, including leaves, shoots, tap roots, pioneer roots, and fibrous roots, were dried and weighed separately. The dried biomass of shoots and roots and their ratio were calculated.

#### 4.7.2. Estimation of *T. melanosporum* Colonization

*T. melanosporum* root colonization was assessed using morphological methods. Five root fragments (5–10 cm long, with all the branching roots) were randomly selected from the proximal and the distal parts of the root system of each plant, with proximal as the roots originating closer to the shoot–root junction, (root collar) and distal as those originating at the end of the main root (tap root). The root colonization with *T. melanosporum* was assessed by analyzing the root tips under a stereomicroscope equipped with a camera (Leica, Wetzlar, Germany). The percentage of ECM colonization was calculated as the ratio of the number of mycorrhizal root tips to the total number of root tips for each plant.

Overall, about 700 root tips were examined for each plant: 300 from the proximal section and 400 from the distal section.

### 4.8. Statistical Analysis

A one-way ANOVA, followed by multiple-comparison tests, was used to compare measured plant characteristics among different treatments (Tukey HSD test, *p* ≤ 0.05). The percentages of *T. melanosporum* mycorrhization were normalized using the arcsine square-root transformation before proceeding to statistical analysis.

## 5. Conclusions

The microbes associated with truffle fruiting bodies play a very important role during the truffle lifecycle, influencing both the plant root affinity with ECM fungi and root architecture and growth, resulting in an increased surface area for nutrient exchange and other rhizosphere effects essential for promoting truffle cultivation. Bacteria can be used to improve seedling quality and mycorrhization in nurseries to promote more successful truffle production. Our results suggest that the *Pseudomonas* sp. inoculum has a potential commercial application as a treatment for truffle-inoculated *Q. ilex* seedlings to improve both seedling quality and mycorrhizal colonization under nursery conditions. The addition of *Pseudomonas* sp. leads to seedlings with a good shoot-to-root ratio, a developed root system, and a higher ECM colonization level compared with the other bacterial formulations. Further studies may be required to explore the mechanisms underlying bacterium and mycorrhiza interaction and to determine their optimal combination for promoting plant and truffle development. Moreover, the performance of the plants in the field and the physiological development of plants over time, as well as the implications for truffle yields, should be explored, too.

## Figures and Tables

**Figure 1 plants-13-00224-f001:**
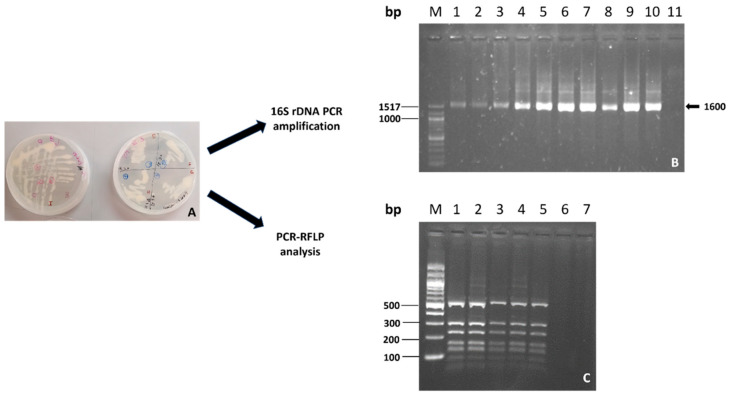
Bacterium identification in the preparatory phase. (**A**) *Bradyrhizobium* growth plates with YM medium. (**B**) PCR amplification of 16S rDNA M Molecular DNA Marker 100 bp DNA Ladder. Lanes 1–4: *Pseudomonas* samples; Lanes 5–9: *Bradyrhizobium* samples; Lane 10: positive control; Lane 11: negative control. (**C**) RFLP analyses of the 16S rDNA of *Bradyrhizobium* M DNA Marker 100 bp DNA Ladder. Lanes 1–5: digestion profile *Rsa*I/16S obtained from 16S amplicons from samplings at different times in the preparatory process.

**Figure 2 plants-13-00224-f002:**
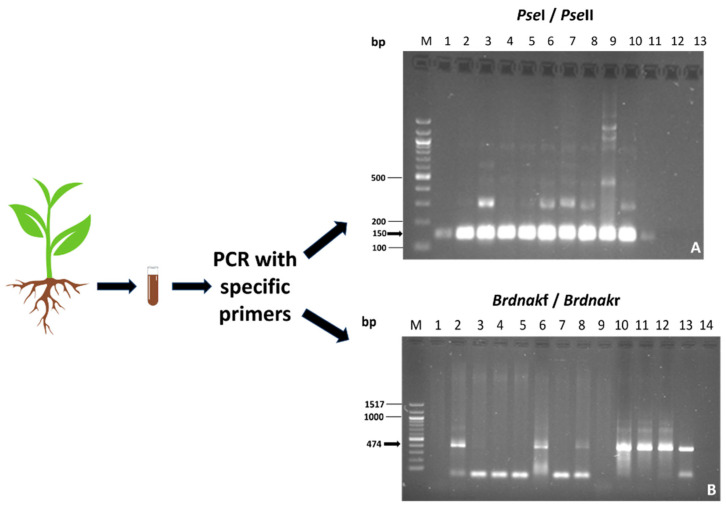
PCR analyses with specific primers. (**A**) Amplification with *Pse*I and *Pse*II of *Pseudomonas*. M DNA Marker 100 bp DNA Ladder; Lanes 1–5: samples from M2 plants; Lanes 6–10: samples from M3 plants; Lane 11: positive control; Lane 13: negative control. (**B**) Amplification with *Brdnak*f and *Brdnak*r primers. M DNA Marker 100 bp DNA Ladder; Lane 1: negative control; Lane 2: positive control; Lanes 3–5: samples from M0 plants; Lanes 6–9: samples from M3 plants; Lanes 10–13: samples from M1 plants. Samplings from mixture-1-treated seedlings were positive for the presence of *Bradyrhizobium*; samplings from mixture-2-treated seedlings were positive for the presence of *Pseudomonas*; samplings from mixture-3-treated seedlings were positive for the presence of both species.

**Figure 3 plants-13-00224-f003:**
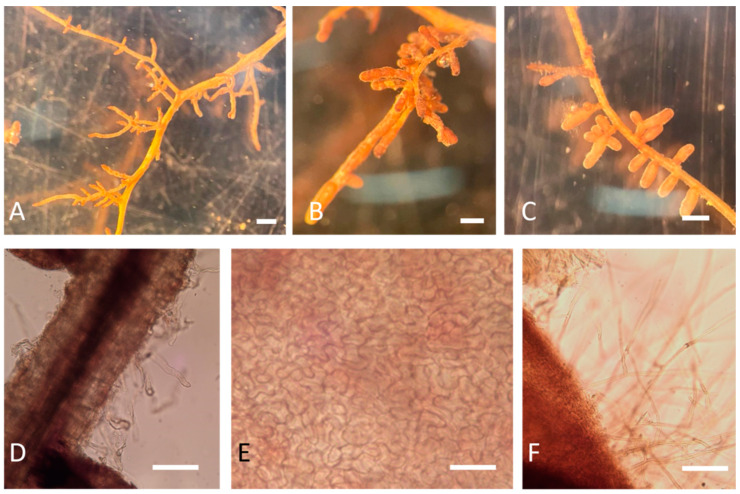
Morphological analyses of *Q. ilex* root tips treated and not treated with *T. melanosporum* under a stereomicroscope (**A**–**C**) and light microscope (**D**–**F**) four months after the treatment. (**A**) *Q. ilex* non-inoculated (M0); (**B**,**C**) *T. melanosporum* mycorrhized root (M2); (**D**) non-mycorrhized root (M0); (**E**) *T. melanosporum* ectomycorrhizal mantle; (**F**) *T. melanosporum* ectomycorrhizal cystidia (scale bars: (**A**–**C**) = 500 µm; (**D**,**F**) = 100 µm; (**E**) = 20 µm).

**Figure 4 plants-13-00224-f004:**
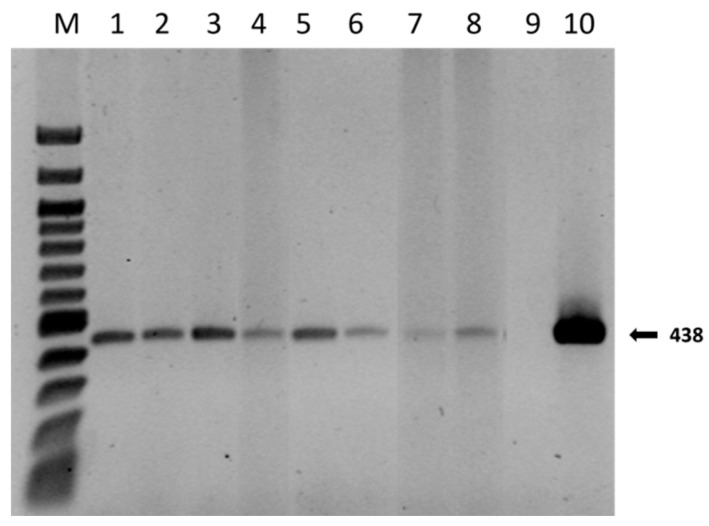
*T. melanosporum* specific DNA amplification from inoculated *Q. ilex* root tips using ITSML and ITSLNG primers, four months after inoculation. M: DNA Marker 100 kb Ladder; Lanes 1–2: root tips from M1-treated plants; Lanes 3–4: root tips from M2-treated plants; Lanes 5–6: root tips from M3-treated plants; Lanes 7–8: root tips from untreated plants (M0); Lane 9: negative control; Lane 10: positive control.

**Figure 5 plants-13-00224-f005:**
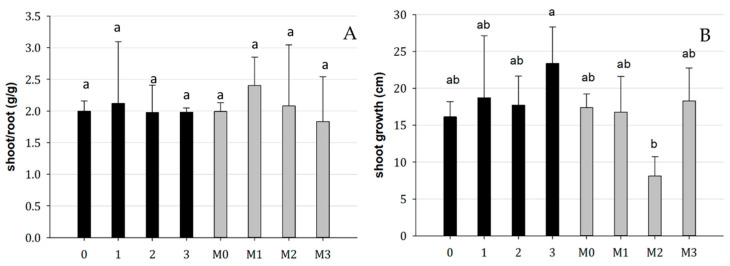
(**A**) Shoot/root biomass ratio and (**B**) shoot growth in mycorrhized (M0, M1, M2, M3) and non-mycorrhized plants, with bacterial inoculum (0: blank, 1: *Bradyrhizobium*, 2: *Pseudomonas*, 3: *Bradyrhizobium* + *Pseudomonas*). Bars indicate standard error; different letters indicate statistical difference for the Tukey HSD test (*p* > 0.05).

**Figure 6 plants-13-00224-f006:**
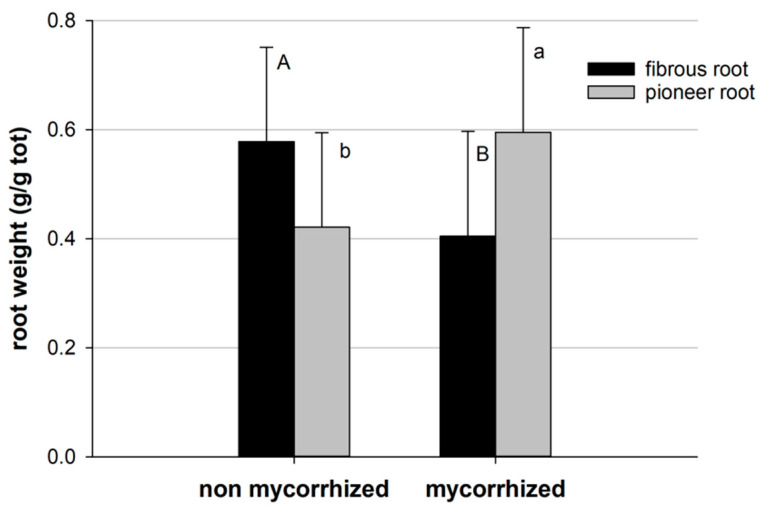
Root biomass of pioneer and fibrous roots in mycorrhized (mean of M0, M1, M2, M3) and non-mycorrhized (mean of 0, 1, 2, 3) plants. Bars indicate standard error; different letters indicate statistical difference between mycorrhized and non-mycorrhized plants (Tukey HSD test, *p* > 0.05).

**Figure 7 plants-13-00224-f007:**
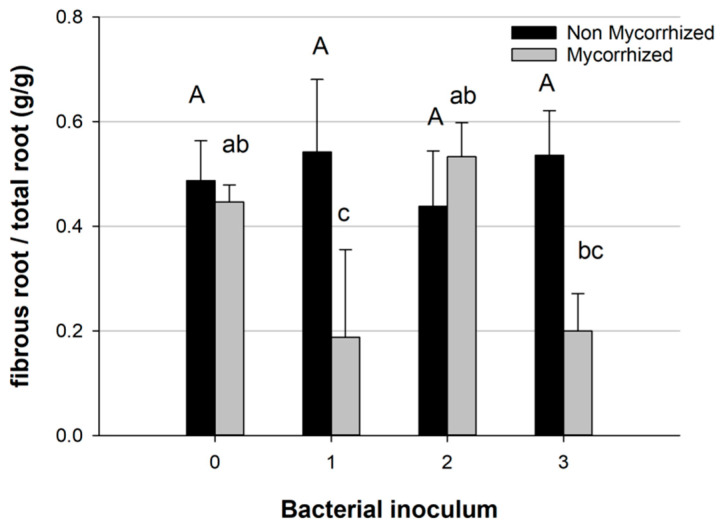
Proportion of fibrous root in total root biomass in mycorrhized and non-mycorrhized plants, with bacterial inoculum (0 blank, 1 *Bradyrhizobium* sp., 2 *Pseudomonas* sp., 3 *Bradyrhizobium* sp. + *Pseudomonas* sp.). Bars indicate standard error; different letters indicate statistical difference between bacterial inocula within mycorrhized and non-mycorrhized plants (Tukey HSD test, *p* > 0.05).

**Figure 8 plants-13-00224-f008:**
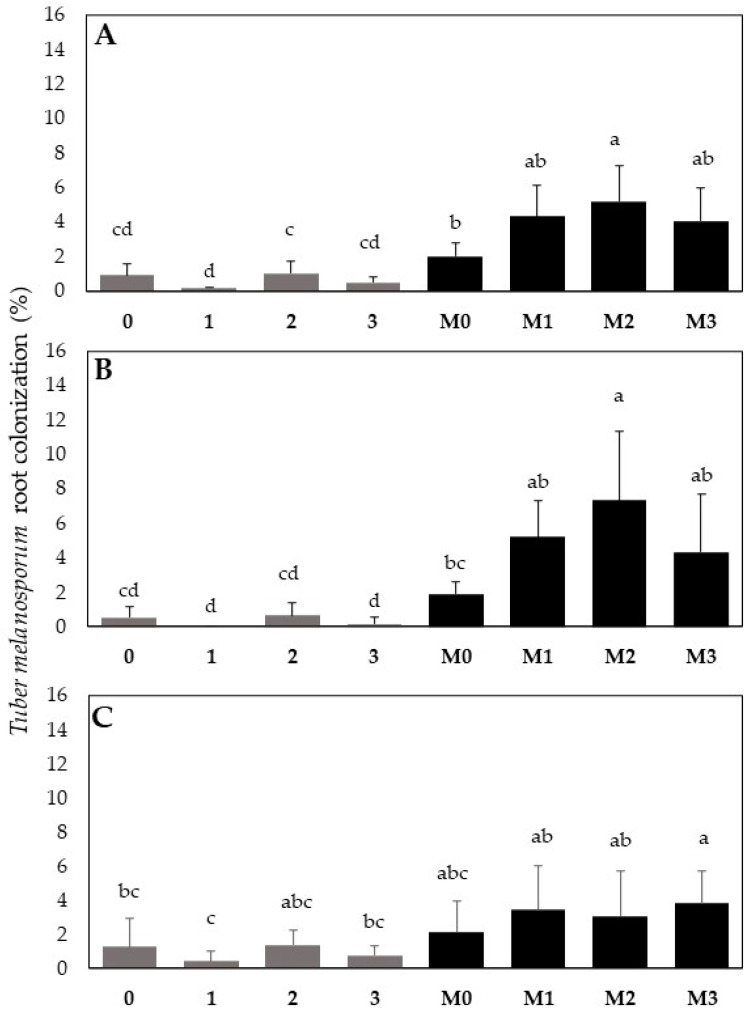
Effect of the different bacterial formulations on *Q. ilex* ECM root tip colonization by *T. melanosporum* observed after six months from fungal inoculation and four months from bacterial co-inoculation, respectively. The *T. melanosporum* ECM colonization rate detected on: (**A**) the total root tips of the root apparatus; (**B**) the root tip proximal to the root–shoot junction (root collar); (**C**) the distal root tips (those originating at the end of tap roots). See the text for more details. Note: plants inoculated with bacteria alone include the following. 0: blank, 1: *Bradyrhizobium* sp., 2: *Pseudomonas* sp., 3: *Bradyrhizobium* sp. + *Pseudomonas* sp. Plants inoculated with both *T. melanosporum* and bacterial mixtures include M0, M1 M2, and M3. Bars indicate standard deviation; different letters indicate statistical difference for the Tukey HSD test (*p* > 0.05).

## Data Availability

The data presented in this study are available in the article.

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
