# Peer review of "Effect of Bacteria Inoculation on Colonization of Roots by Tuber melanosporum and Growth of Quercus ilex Seedlings"

_plants, 2024, doi:10.3390/plants13020224_

Round 1
Reviewer 1 Report
Comments and Suggestions for Authors
The paper is devoted to the economically important topic. The study is well designed, the methodology applied is appropriate. Nevertheless, there are several aspects in the text, which should be corrected and improved.
Lines 108-114: This section contains the methodological description and does not belong to Results.
The authors use the abbreviation "spp." through the text. "spp." means species in plural. Does it mean that several Bradyrhizobium and Pseudomonas species were used in the formulations, or it was one Bradyrhizobium sp. and one Pseudomonas sp., which were not identified to species level?
Subsection 4.3: how was the density of UFC (the abbreviation should be explained) calculated?
Line 318: "The seedlings were arranged in a complete random design" – what does it mean? How is the completely random design applicable to such kinds of study?
Lines 396-397: "morphological root system analysis" – what does it mean? Should be explained.
Subsection 4.7.1. Results presents not only the measurements of shoot and root growth, but also biomass, which is not mentioned here. At the same time, Results does not present the mentioned here data on number of leaves.
Subsection 4.7.2. Why to count the number of non-mycorrhized root tips, if the colonization rate was calculated by dividing the number of mycorrhized root tips by the total number of root tips? "closer" and "further" are non-concrete terms, the distance from shoot-root junction should be defined.
Discussion contains unclear and awkward sentences and statements (see the attached PDF file), which should be explained and rewritten.
All other comments, corrections, and suggestions are inserted into the attached PDF file of the manuscript.

Comments on the Quality of English LanguageAuthor Response
We would like to thank reviewer 1 for suggesting improvements to the manuscript. We have incorporated the
suggested changes which are visible in the text in red.
Here we report the explanations of specific points requested in the letter.
Our answers are highlighted in bold type. The lines correspond to the revised manuscript.
- Lines 108-114: This section contains the methodological description and does not belong to Results.
The sentences have been deleted.
- The authors use the abbreviation "spp." throughout the text. "spp." means species in the plural. Does
it mean that several Bradyrhizobium and Pseudomonas species were used in the formulations, or it was
one Bradyrhizobium sp. and one Pseudomonas sp., which were not identified to species level?
"Spp." was replaced with "sp.". The species were not identified.
- Subsection 4.3: how was the density of UFC (the abbreviation should be explained) calculated?
The abbreviation and the calculation were explained in the text.
- Line 318: "The seedlings were arranged in a complete random design" – what does it mean? How is
the completely random design applicable to such kinds of study?
The paragraph 4.4 was changed, and the sentence deleted. It was indeed not possible a complete
randomization of the treatments, to avoid contamination of the inocula between the plants.
- Lines 396-397: "morphological root system analysis" – what does it mean? Should be explained.
The sentence was changed to “architectural analyses of root system”.
- Subsection 4.7.1. Results presents not only the measurements of shoot and root growth, but also
biomass, which is not mentioned here. At the same time, Results does not present the mentioned here
data on number of leaves.
The text has been modified: only measurements shown in results have been described (biomass of
the different plant parts).
- Subsection 4.7.2. Why to count the number of non-mycorrhized root tips, if the colonization rate was
calculated by dividing the number of mycorrhized root tips by the total number of root tips? "closer"
and "further" are non-concrete terms, the distance from shoot-root junction should be defined.
The subsection 4.7.1 was rewritten to clarify this point.
- Subsection 4.8 was rewritten
- Discussion contains unclear and awkward sentences and statements (see the attached PDF file), which
should be explained and rewritten.
We have reorganized some sentences to make this section clearer.
As mentioned above, all the changes suggested in the PDF text have been made which are in red. We also
noticed further inaccuracies, which we have corrected. The main fixes are listed below:
- Line 42: we replaced “tubers”, which means potato, with “the genus Tuber”.
- Line 43: we changed “for their commercial value” to “to support the production”.
- Line 44: we replaced “biological aspects” with “biology”.
- Lines 52-53: we added “Different species can be used as host plants, the most common are oak species
(Quercus spp.) and among them Quercus ilex is one of the most common species in Mediterranean
area”. We added the reference [7].
- Lines 76-79: we added “colon” to justify the following list.
- Lines 92-93: the sentence was improved.
- Subsection 2.1, lines 112-113: the sentence was rewritten.
- Figure 3 was improved with scale bars, and the legend was implemented with the meanings.
- Line 159: we added “caused”.
- Figure 5 was implemented with the significant letters in part A.
- Lines 179-180: “allocation” was changed with “biomass”.
- Figure 7: the caption was changed to “proportion of fibrous root on total root biomass...”.
- Lines 200-201: the sentence was rewritten.
- Lines 214-216: the sentence was rewritten.
- Line 235: “between Q. ilex and T. melanosporum” was added for more clarity.
- Line 240: the sentence was rewritten.
- Lines 251-254: the sentences were rewritten.
- Lines 258-263: the sentences were rewritten.
- Lines 271-275: the sentences were rewritten.
- Line 292: we corrected with “during transplantation of seedlings in pots”.
- Subsection 4.3 was changed to better explain the method.
- Line 324: the sentence was deleted and subsection 4.4 was modified for more clarity
- Lines 334-335: plants disposition in the greenhouse was added
- Line 337: the sentence was rewritten.
- Subsection 4.7 was modified.
- Lines 423-426: the sentences were rewritten.
- Subsection 4.8 was rewritten.
- The bibliography has been updated with the addition of reference 7.
We hope that everything is clear and that we can proceed with publication. We are available for any further
change.
Best regards
Antonella Amicucci

Reviewer 2 Report
Comments and Suggestions for Authors
As the authors state themselves the real influence of root colonization by tested bacteria on tuber growth would greatly increase the interest of this work.
Author Response
We have improved the manuscript rewriting some sentences that are visible in the text in red.
We thank reviewer 2 for appreciating our research.
Best regards